# High-frequency recombination between members of an LTR retrotransposon family during transposition bursts

Diego H. Sanchez[1], Hervé Gaubert[1], Hajk-Georg Drost[1], Nicolae Radu Zabet[1,2] & Jerzy Paszkowski[1]

Retrotransposons containing long terminal repeats (LTRs) form a substantial fraction of eukaryotic genomes. The timing of past transposition can be estimated by quantifying the accumulation of mutations in initially identical LTRs. This way, retrotransposons are divided into young, potentially mobile elements, and old that moved thousands or even millions of years ago. Both types are found within a single retrotransposon family and it is assumed that the old members will remain immobile and degenerate further. Here, we provide evidence in Arabidopsis that old members enter into replication/transposition cycles through high rates of intra-family recombination. The recombination occurs pairwise, resembling the formation of recombinant retroviruses. Thus, each transposition burst generates a novel progeny population of chromosomally integrated LTR retrotransposons consisting of pairwise recombination products produced in a process comparable the sexual exchange of genetic information. Our observations provide an explanation for the reported high rates of sequence diversification in retrotransposons.

[1] The Sainsbury Laboratory, University of Cambridge, Cambridge, CB2 1LR, UK. [2] Present address: School of Biological Sciences, University of Essex, Colchester, CO4 3SQ, UK. Diego H. Sanchez and Hervé Gaubert contributed equally to this work. Correspondence and requests for materials should be addressed to J.P. (email: jerzy.paszkowski@slcu.cam.ac.uk)

Transposons change chromosomal position and copy number and are thus a highly dynamic part of eukaryotic genomes. There are two main classes of transposons: DNA elements that move by a cut-and-paste mechanism and retrotransposons that replicate their DNA copies through an RNA intermediate. Retrotransposons are further subdivided into those containing and those lacking long terminal repeats (LTRs). The transposition cycle of LTR retrotransposons, which are the major component of plant genomes[1], starts with transcription initiation within the 5′ LTR. The transcript, which extends almost over the entire length of a retrotransposon, encodes a structural GAG protein and a polyprotein. The latter is processed by a retrotransposon-encoded protease into reverse transcriptase, RNAseH, and an integrase. These, together with a subset of transcripts, are packaged into GAG-derived virus-like particles (VLPs)[2]. Subsequent reverse transcription involves two transfers of DNA strands that lead to the synthesis of a complete copy of a retrotransposon as extrachromosomal DNA (ecDNA) terminated by two identical LTRs (Supplementary Fig. 1). Integrase activity inserts ecDNA into host chromosomes and results in new chromosomal copies.

With time, newly integrated copies accumulate mutations that randomly diversify their sequences. Diverging copies of related retrotransposons evolve into families whose members differ increasingly in DNA sequence. The differences in the sequences of initially identical LTRs can be used to estimate the time of the last transposition event[3]. Using this notation, young family members that transposed recently are considered to be potentially active and old members to represent historic transposition events. It has been shown that particular steps of the transposition cycle of LTR retrotransposons, such as reverse transcription and chromosomal integration of DNA, are analogous to the infection cycles of retroviruses[4]. However, in contrast to the well-documented formation of recombinant strains during retroviral mix-infections[5], the genetic consequences of the transposition bursts of an entire family of LTR retroelements, consisting of young and old members, are unknown. Here, we examined such consequences using an experimentally induced retrotransposition burst of the endogenous, heat stress-activated family of *Arabidopsis* LTR retrotransposons *Copia78* or *Onsen*[6].

## Results

**Recombination between members of Onsen family.** *Onsen* family in ecotype Col-0 comprises eight members, which differ in DNA sequence and the level of divergence between the two LTRs, i.e., in age (Fig. 1a). Three members (AT1G11265, AT3G61330, and AT5G13205, referred to and labelled in all figures as "red", "yellow", and "grey", respectively) have identical LTRs, indicative of recent transposition and also of the possible retention of transpositional activity[6,7]. The remaining five family members are old and have accumulated mutations in their LTRs, reducing sequence identity between the 5′ and 3′ LTRs (AT1G48710 "violet" and AT3G59720 "blue"—to 99%; AT3G32415 "orange"—98%; AT1G21945 "white" and AT1G58140 "green"—97%) (Fig. 1a). This degree of LTR divergence indicates that the last movements occurred 0.5–1 million years ago[3]. In addition, three out of five old members ("white", "orange", and "blue") have alterations in the open reading frames of protein-coding regions, and thus their possible transposition would presumably involve proteins supplied by other members of the *Onsen* family[6].

The *Onsen* family accumulates transcripts and ecDNA in heat-stressed wild-type plants. However, none of the members transpose under these conditions[6]. In contrast, transposition results in new chromosomal insertions in the progeny of heat-stressed mutant plants deficient in the biogenesis of small

interfering RNAs (siRNAs). An example is the *nrpd1-3* strain mutated in a gene encoding plant-specific RNA polymerase IV required for siRNA production[6]. However, it is not clear which members of the *Onsen* family are transposition competent. In a scenario favouring the activity of young members, the obvious expectation is that all or some of the three youngest members ("red", "yellow", and "grey") are able to move.

To test this assumption, we first examined transcript levels of the *Onsen* family in heat-stressed wild-type and *nrpd1-3* mutant plants and confirmed heat stress-triggered accumulation (Supplementary Fig. 2a)[6]. Subsequently, we determined the transcript levels of each member of the family (Fig. 1b). For this we counted the number of NGS reads of RNA-seq after assigning them to each *Onsen* member using three independent regions of sequence polymorphisms characteristic of each member. Since "red" and "yellow" are highly homologous, their reads were first pooled using three generally polymorphic regions and then divided between each member using one specific distinguishing polymorphism. These RNA analyses showed indeed that young family members respond to heat stress with the highest levels of transcripts, especially the *nrpd1-3* mutant (Fig. 1b). However, transcripts of the two older members "violet" and "blue" also contributed significantly to the family mRNA pool. Interestingly, "green" showed a very low transcript level in wild-type but was highly activated in the *nrpd1-3* mutant (Fig. 1b). Transcript levels of the remaining older members "white" and "orange" remained very low (Fig. 1b).

From whole-genome sequencing data, we isolated *Onsen*-specific pair-end reads derived from PCR-free NGS of DNA from wild-type and *nrpd1-3* plants sampled under control conditions or directly after heat stress. The relative numbers of *Onsen*-specific reads increased by heat stress threefold in the wild-type and 30-fold in *nrpd1-3* (Supplementary Fig. 2b). These data confirmed previous reports that heat stress induces the accumulation of *Onsen* ecDNA in both genotypes[6]. Subsequently, we determined the abundance of ecDNA for each family member by reads distribution analogous to that for RNA-seq. The ecDNA levels corresponded roughly to transcripts levels except for the "white" member, for which no ecDNA was detected. Interestingly, levels of ecDNA in the *nrpd1-3* strain of the "green" member, one of the oldest, were similar to the youngest members of the *Onsen* family (Fig. 1c).

To determine the chromosomal integration proficiency of different family members, we sequenced the DNA of the pooled progeny of a heat-stressed *nrpd1-3* plant in which we had previously detected 75 new *Onsen* insertions[8]. We first developed a customised bioinformatics workflow to identify the parental origin of new insertions. By anchoring one of the reads to a junction between each newly inserted LTR and the chromosomal DNA, this workflow used additional paired reads in these regions to reconstruct the LTR sequences of each newly integrated *Onsen* copy. Due to the similarity of LTR sequences between members of the *Onsen* family, unambiguous assignment to a particular member was only possible for 22 newly inserted copies (Fig. 2). These included 9 LTRs of the young "grey" member (Fig. 2, numbers 11–16 and 51–53) as well as, surprisingly, 13 LTR insertions of old members, namely, 3 insertions of the "violet" element (Fig. 2, numbers 5, 27, 32), and 10 insertions of the "green" element (Fig. 2, numbers 28, 29, 33, 34, 45–49). Explicit parental assignment was not possible for a further 32 new insertions. However, 31 of these involved the remaining young members "red" and "yellow" (Fig. 2, numbers 18–21 and 54–75) and 1 may have been the result of a movement of the older "blue" member (Fig. 2, number 17). Most astonishingly, the remaining 21 new insertions contained novel LTRs derived from more than one *Onsen* member. Given that we used a genome sequencing

method that excludes a PCR amplification step, we concluded that the newly inserted hybrid LTRs are most likely intra-family recombination products. They consisted of sequence pairs derived from all *Onsen* family members except "white" (Fig. 2), for which we had previously found no ecDNA (Fig. 1c).

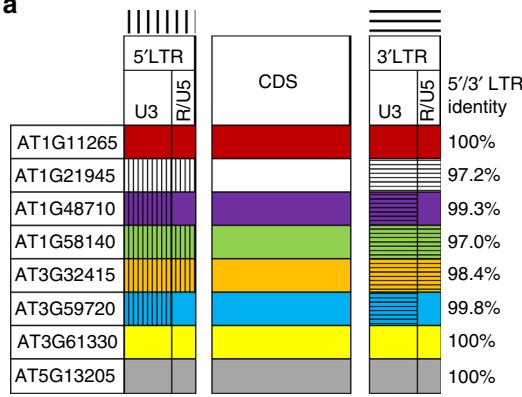

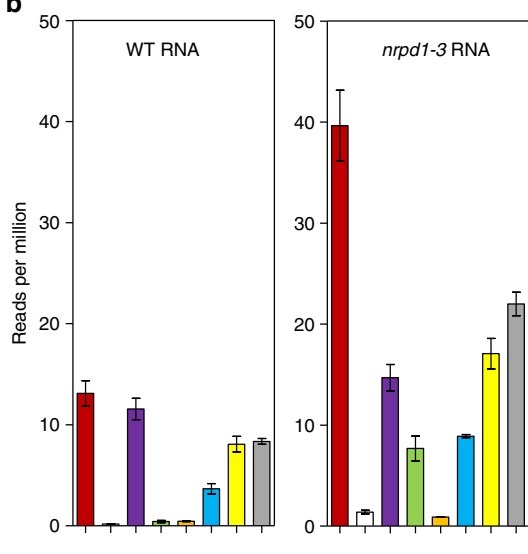

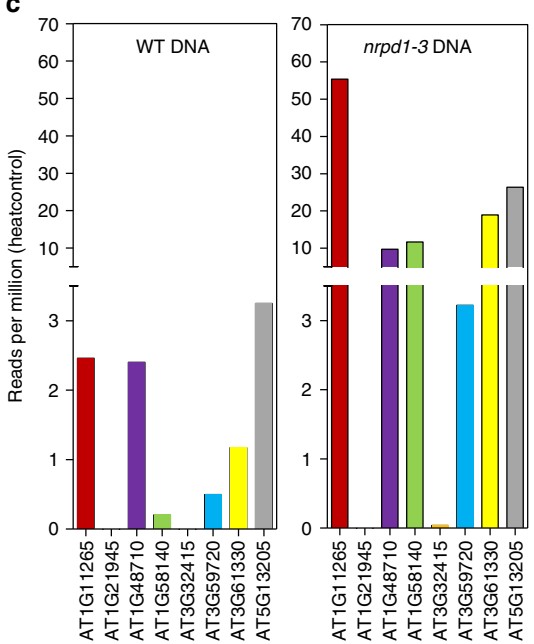

**Fig. 1** Heat stress activation of individual *Onsen* members at both the transcript and ecDNA levels. **a** Colour key for each *Onsen* member; retrotransposon domains such as LTRs (with U3 and R/U5 regions) and central coding-sequences (CDS) are depicted. Percent identities of 5'/3' LTRs of each member are displayed at the right. In those older members in which 5' and 3' LTRs differ, vertical (5') and horizontal (3') lines depict distinct sequences for different LTRs. Old members AT1G48710/"violet" and AT1G58140/"green" display complete central CDS. **b** Heat-induced transcript levels for each individual *Onsen* member derived from RNA-seq analysis of wild-type and *nrpd1-3* plants subjected to heat stress. Data represent means ± SEM for *n* = 2 independent biological replicates. **c** Heat-induced ecDNA abundance for each individual *Onsen* member, as inferred from whole-genome sequencing of wild-type and *nrpd1-3* plants under control growth or subjected to heat stress. The bars represent subtraction of the control counts from the heat sample counts. In both **b**, **c**, NGS reads mapping to the *Onsen* family were assigned to individual members using unambiguous sequences matching each individual member in independent areas (further details in the text) and then counted, normalised to mapped library size, and averaged. Colour code as in Fig. 1a

Additional reconstruction of LTR sequences from new *Onsen* insertions in an independent *nrpd1-3* plant produced comparable results (Supplementary Fig. 3).

To further characterise the parental origin of newly inserted copies of *Onsen*, we determined the complete individual sequences of 32 randomly chosen new insertions (Fig. 3). We designed PCR primers recognising chromosomal sequences flanking each insertion and performed high fidelity PCR to rescue each of them separately. All PCR products were then cloned and subjected to Sanger sequencing. The composition of the LTRs predicted by whole-genome sequencing (Fig. 2) was confirmed for each individual insertion and this was further refined to reduce the number of possible parental origins (Fig. 3). Most recombination products within LTRs were consistent with recombination events occurring during the first DNA template switch (Supplementary Fig. 1), which uses LTR homologies within the R region (Fig. 3, numbers 1–9, 23–26, and 30–31). However, we also identified one LTR recombination event within the U3 region (Fig. 3, number 10). In addition, we detected many recombination junctions in the protein-coding segment of *Onsen* (Fig. 3, numbers 1–11, 14, 17–22, 26–27, and 29–32). Importantly, all but one of the newly formed recombinant *Onsen* copies consisted of sequences derived from just two parental members of the family, suggesting that recombination occurred pairwise. The single exception was new copy number 1 (Fig. 3), which appeared to incorporate sequences of three different parental members, maybe as the result of rare extrachromosomal recombination between ecDNAs. Remarkably, only 5 of these 32 new insertions had a single parent origin (Fig. 3, numbers 12, 13, 15, 16, and 28). Thus, the frequency of recombination appears to be extremely high and a likely explanation for new *Onsen* copies of single parent origin would be that the two co-packaged RNA molecules came from the same parental *Onsen* member. Noticeably, sequences derived from young parental members were found in 30 new insertions, suggesting that young members drive the transposition burst of this family. However, in the recombination process, sequences of old family members were habitually incorporated into progeny insertions (as revealed by analysis of the relative contributions of parental *Onsen* family members to newly integrated copies; Supplementary Fig. 4). This created recombinational diversity among the progeny of the *Onsen* family.

To address the fidelity of replicative transposition, we examined random sequence changes in the sequences of the 32 randomly chosen new inserts. In approximately 160 kb of sequence assigned to new insertions, we detected 10 sequence polymorphisms not present in the parental members (Supplementary Table 1). These

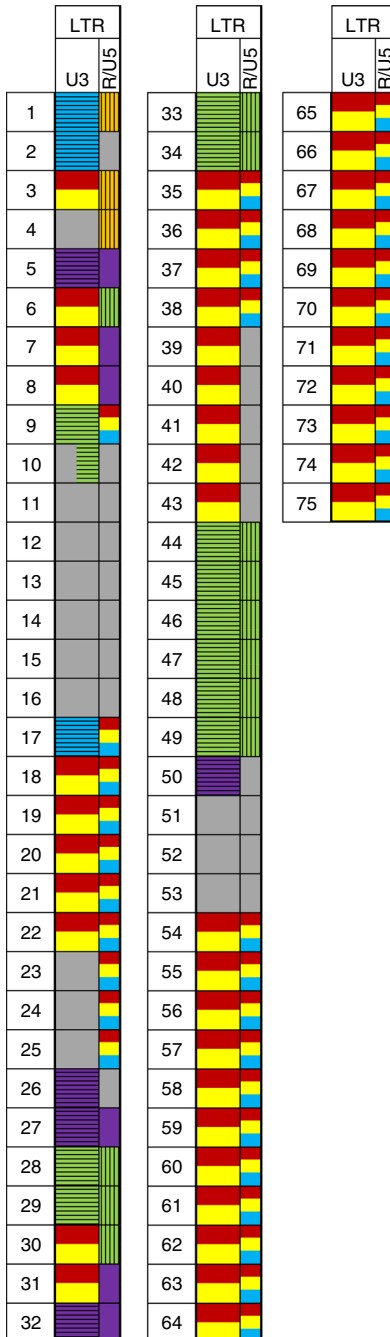

**Fig. 2** Origin of reconstructed LTRs of 75 new chromosomal *Onsen* insertions recovered in the progeny of a heat-treated *nrpd1-3* plant[8]. LTR domains are marked as U3 and R/U5. Informative SNPs and indels were used to infer the parental origin of LTRs. Several colours are used in the same area when polymorphisms were shared between more than one member. Colour code as in Fig. 1a

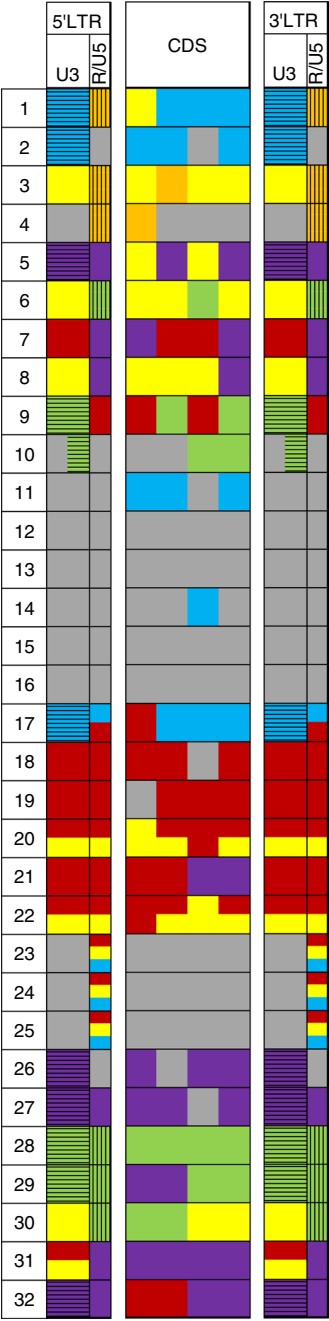

**Fig. 3** Thirty-two cloned and Sanger-sequenced new chromosomal *Onsen* insertions found in progeny of a heat-stressed *nrpd1-3* mutant plant[8]. Informative SNPs and indels were used to infer the parental origin of new inserted retrotransposon copies. Use of the entire sequence of an element minimises the number of possible parental origins and provides a solution for the structure of the recombinant progeny element. Retrotransposon domains and colour code as in Fig. 1a

most likely represent errors during reverse transcription, which are a known source of novel mutations in retroelements[9]. However, this sequence variability is rather marginal compared to the observed sequence diversity derived from the recombination step during a retrotransposition burst, which reshuffles mutations accumulated during years of passive maintenance of retroelements within the host chromosomes.

**Presence of recombinant LTRs within *Onsen* ecDNA.** In almost all cases, recombination appears to have occurred pairwise,

suggesting a transcript pairing rule analogous to retroviruses[5], which has also been postulated for LTR-retrotransposons[2]. Thus, it can be assumed that recombination occurs during reverse transcription of the two co-packaged *Onsen* transcripts. Consequently, ecDNA as a direct product of reverse transcription should consist of recombinant molecules of merged parental origin. To test this prediction, we used the whole-genome sequencing data and examined sequences of ecDNA synthesised directly after heat stress-triggered induction of the *Onsen* replicative cycle. As the average length of reads was 150 bp, we created a virtual library of 501491

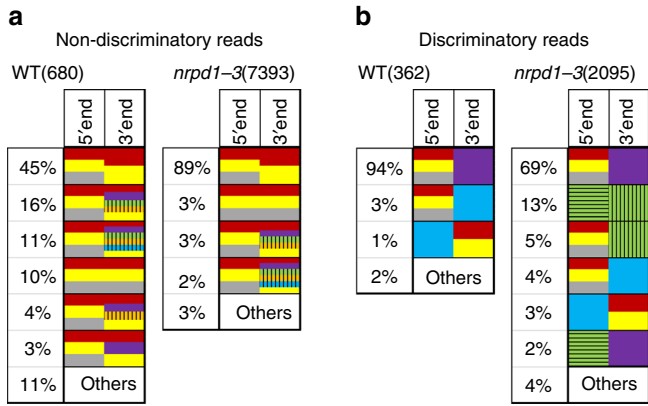

**Fig. 4** DNA sequencing of pair-end reads starting at the ends of LTRs of *Onsen* reveals recombinant LTRs in ecDNA of heat-stressed wild-type and *nrpd1-3* plants. **a** Non-discriminatory pair-end reads (see text). **b** Discriminatory pair-end reads reveal the occurrence of recombinant LTRs. Colour code as in Fig. 1a. Horizontal or vertical bars are present only in those cases where NGS reads matching old *Onsen* members contained polymorphisms that allowed discrimination of 5′ and 3′ LTRs

DNA fragments of 145 bp representing all possible recombination junctions between the eight *Onsen* members (see Methods). Since these particular sequences are not present in the *Arabidopsis* genome without a burst of *Onsen* activity, we were able to search our sequencing libraries from heat-stressed plants for the appearance of sequences indicative of recombination events. Probabilistic and permutation models predicted that approximately 0.6–0.8 recombination events per thousand reads mapping to *Onsen* may be due to random errors of NGS present in our data sets (see Methods). Application of the search strategy of the virtual library to the actual sequencing data gave frequencies of putative recombination junctions below the values expected in samples from control plants (see Methods). In contrast, the frequencies of sequences indicative of recombination were significantly elevated in heat-stressed plants. We mapped 7.37 and 7.55 NGS reads matching recombination products in the wild-type and *nrpd1-3*, respectively, normalised per thousand reads mapping to *Onsen*. We conclude, therefore, that recombination events between *Onsen* members are present in ecDNA from both wild-type and *nrpd1-3* plants subjected to heat stress. Interestingly, the relative frequencies of recombinant reads were similar in wild-type and *nrpd1-3*, implying that their formation is not influenced by the nrpd1-3 mutation.

We designed a second approach to verify the presence of recombinant molecules within *Onsen* ecDNAs. We examined the composition of perfect-matching pair reads in which the sequence of one mate starts exactly at the beginning of the LTR, which is diagnostic of linear ecDNA. We recovered 1042 and 9488 read pairs in DNA of wild-type and *nrpd1-3* heat-stressed plants, respectively. Subsequently, we assigned the 5′ read and 3′ read of each read pair to a member of the *Onsen* family (Fig. 4). A high proportion of reads could not be assigned unambiguously to a particular *Onsen* member, due to the sequence similarities between members. Thus, the origin of ecDNA could not be resolved unequivocally in these cases. Such reads were termed "non-discriminatory", although the presence of recombinant ecDNA among them cannot be excluded entirely (Fig. 4a). In contrast, the "discriminatory" reads clearly indicated the presence within ecDNA of recombinant LTRs derived from two different members of the *Onsen* family (Fig. 4b). Notably, almost all "discriminatory" reads in wild-type suggested efficient recombination between the youngest family members with the older "violet" or "blue" members; however, a contribution of the oldest "green" member to recombinant LTRs was not detected. In

*nrpd1-3*, where transcripts of the "green" member were much more abundant (Fig. 1b), this parental member also contributed significantly to the extrachromosomal recombinant LTRs we detected (Fig. 4b).

The two complementary analyses of LTR sequences synthesised directly after heat stress activation of *Onsen* family members have provided evidence of recombination products within *Onsen* ecDNA. Therefore, recombination most likely takes place during reverse transcription of the two RNA molecules packaged into VLPs. This concept is consistent with the pairwise parental origin of progeny recombinant copies newly inserted into host chromosomes (Fig. 3).

## Discussion

It has long been recognised that TEs diversify genetically at a greater pace than protein-coding genes[10], but it was not clear how this rapid genetic diversity arises. Putative recombination events have been inferred from phylogenetic analysis of yeast, *Drosophila* and plant elements[11–14], which implies that historical recombination of related LTR retrotransposons occurred in these evolutionary distant hosts. In addition, the widely documented recombination of retroviruses[5,15,16] has often been extrapolated to LTR retrotransposons. However, the only experimental support so far has come from the transgenic supply of marked yeast Ty elements[17,18] or the accidental recovery of a putative recombination product, where PCR-derived artefacts during standard NGS sequencing could not be ruled out[19]. Here, we provide direct in vivo evidence of high-frequency intra-family recombination of LTR retrotransposons as natural inhabitants of the *Arabidopsis* genome. We have demonstrated high-frequency recombination during a retrotransposition burst within a single plant generation that resulted in complex reshuffling of parental sequences. In this way, a novel population of chromosomal retrotransposon copies can arise with unprecedented speed.

Like retroviruses, LTR retrotransposons also have a "pseudo-diploid" phase in their life cycle involving two molecules of parental genomic RNA that contribute to the formation of recombinant progeny in the form of ecDNA[5]. This process resembles (pseudo)sexual reproduction and is likely to be advantageous for accelerated molecular evolution and enhanced genetic diversity, thus contributing to the survival of retrotransposon families[20]. Most family members are involved in this process and the composition and transposon sequences of families are likely to be different in distantly related individuals of a given species. This was exemplified recently by the transposon diversity found among *Arabidopsis* accessions[21]. Thus, it can be envisaged that outcrossing of a host further increases the genetic diversity of LTR retrotransposons in the hybrid progeny and that this has profound effects on burst-driven diversity of recombination products.

It can also be foreseen that the structure of the progeny family resulting from a transposition burst will relate directly to the composition of the parental members and the ability of particular members to contribute to the process. It may be that the prevalent involvement of young, autonomous elements "resurrects" older relatives. In contrast, should old degenerated elements dominate the burst, this would result in a rapid decrease in fitness of the descendants and accelerate extinction of that particular family of elements. Thus, the high frequency of recombination of LTR retrotransposons described here during their consolidated movement may profoundly influence the structure of such retrotransposon families and inevitably also of host chromosomes.

## Methods

**Plant material and experimental conditions**. *A. thaliana* Col-0 seeds were surface sterilised and sown in petri dishes with 0.5× MS medium (Duchefa), containing 1%

sucrose, 0.05% MES at pH 5.7, and 0.8% w/w of Bacto Agar (Becton Dickinson). After 3 days of stratification at 4 °C, plants were grown in a CU-22L growth chamber (Percival) under 12/12 h (day/night) light cycle at 21 °C. For the heat induction of *Onsen*, 7 days old plantlets were first placed for 24 h on top of a chilling platform within the growth chamber and maintained by an external mini-chiller at 4 °C, with the chamber temperature at 6 °C. This treatment was followed by 24 h at 37 °C for the heat-stressed plants, whereas control plants were moved back to 21 °C. Pools of 50–70 seedlings from independent biological replicates were collected directly after heat stress and control treatments. The *nrpd1-3* mutant[22], and the *nrpd1* plant 2L and *nrpd1* plant 5L2 with known positions of 75 and 20 new *Onsen* insertions, respectively, were characterised previously[8].

**RNA-seq and DNA-seq.** For transcriptome and whole-genome sequencing analyses, 50–70 seedlings were pooled for RNA and DNA extraction. Total RNA and DNA were isolated using the Plant-RNAeasy kit (Invitrogen) or the DNEasy Plant Mini Kit (Qiagen), respectively. Strand-specific libraries for RNA-seq were prepared with 2 µg of RNA using the TruSeq Stranded mRNA Library Prep Kit (Illumina), whereas PCR-free libraries for the whole genome sequencing were prepared with 1 µg of DNA using the TruSeq DNA PCR-Free Library Prep kit (Illumina); in both cases following the provider's instructions. Sequencing was performed in a Next-Seq 500 platform (Illumina) reporting 150 bp pair-end reads. Each sample was sequenced to at least an average depth of 36× fold.

New *Onsen* insertions were PCR-cloned individually using Phusion® High-Fidelity DNA Polymerase (NEB) and primers placed in flanking genomic DNA. PCR products were cloned with CloneJET PCR cloning Kit (Thermo Fisher) and Sanger-sequenced with multiple primers designed over *Onsen* sequences shared between members. All primers used for the initial experiments revealing *Onsen* recombination (displayed on Supplementary Fig. 3b) and for the subsequent experiments displayed on Fig. 3 are available in Supplementary Table 2.

**Data handling and analysis.** NGS data was trimmed using Trimmomatic[23] with the parameter sets TruSeq2-PE.fa:2:10:5:1 LEADING:20 TRAILING:20 SLIDINGWINDOW:4:15 MINLEN:36; and subsequently mapped to TAIR10 *A. thaliana* assembly. DNA sequencing was mapped with Bowtie2[24] using parameter sets --very-sensitive -X 1000 --non-deterministic. RNA-seq was mapped with TopHat[25], with parameter sets --max-multihits 1 --read-realign-edit-dist 0 --no-mixed and using a Bowtie2 index. Open-source software such as SAMtools[26] and Picard (http://picard.sourceforge.net) were applied in subsequent handling of NGS data. *Onsen*-mapping reads were recovered with no library deduplication. Several custom-made workflows were developed in-house for various data manipulations and analysis, using SAMtools[26], BEDtools[27], and Python scripts (www.python.org). These are available at https://github.com/diegohernansanchez/.

For revealing chromosomal coordinates of new *Onsen* insertions in genome sequencing from progeny of heat-activated *nrpd1-3* background, a combined strategy based on recovering "discordant paired-end" reads and "junction" reads around insertion points was used. First, an *Onsen*-masked genome was used to map pair-end reads, recovering those in which one mate from the pair mapped to a chromosomal location but the other remained unmapped, and then filtering for those in which the second mate mapped to *Onsen*'s LTR. The recovered discordant paired-end reads putatively reflect new insertions, but also non-discordant pair-end reads mapping to *Onsen* members, which were used as positive controls for this approach. In a second step, we recognised unmapped reads that blasted to *Onsen*'s LTR 5′ and 3′ extremities and trimmed away the transposon sequence; re-mapping these short sequences to the genome using "soft clipped" Bowtie2 mapping (to account for tandem-site-duplication generated by the new insertions) with parameters sets --local --very-sensitive-local --score-min L,5,0 --np 0. This approach recovered all the junction reads between the genome and *Onsen*. Chromosomal mapped reads from the first and second steps clearly accumulated around *Onsen* members and putative new *Onsen* insertion points. Finally, a intersect between the first and second steps along with manual assessment was used to confidently define new insertion's coordinates, comparing the sequenced *nrpd1-3* and wild-type backgrounds to rule out false positives. This strategy has been previously validated[8]. The sequencing for finding new insertions was performed on DNA isolated from control progeny of plants not subjected to heat stress, and therefore ecDNA was absent.

For reconstructing LTRs of new insertions by genome sequencing, we retrieved pair-end reads anchoring around the chromosomal insertion point (recovered from the first step described above for finding chromosomal coordinates of new *Onsen* insertions). The sequences of the second mate of the pair mapping to *Onsen*—from both the 5′ and 3′ edges of the new insertion—were re-assembled and aligned to known *Onsen* LTRs using the commercial Geneious software (www.geneious.com). The validity of this strategy was confirmed with the cloning and Sanger sequencing of 32 new insertions; from these the LTR sequences were identical to the reconstructed LTRs (Fig. 3).

For *Onsen* member-specific heat responsiveness at transcript and ecDNA level, we first recovered all *Onsen* mapping reads. Then, we counted the number of reads after unambiguous assignment to each individual member, using the arithmetic mean of three characteristic 70 bp sequence "addresses" (displayed in Supplementary Table 3) anchored in independent regions of sequence polymorphisms. The assignment was made by perfect string matching of these addresses. The validation of this approach for estimating transcript levels was

corroborated by correlation with standard transcript level determination (RPKM), across both wild-type transcriptome data and case examples of housekeeping and heat responsive genes (Supplementary Fig. 5).

**Assessment of *Onsen* member contributions to new insertions.** To assess the relative contributions of each *Onsen* family member to the sequences of the 32 new insertions, we devised a scoring system that quantified each recombination event in terms of the contribution of a particular parental *Onsen* to the new recombined insertion. We assigned up to 12 points per new insertion. Where only one parental member contributed to a new insertion, we assigned 12 points. We assigned each member six points where two different parental family members formed a new insertion through a recombination event; where three members contributed, each received a score of four points, and so on. We then calculated the total "contribution scores" for each *Onsen* member over all 32 new insertions (32 scores per member). Finally, we calculated and visualised the relative frequencies of these total contribution scores to determine whether they matched the relative levels of transcripts assigned to each parental *Onsen* (Supplementary Fig. 4).

**Search for recombination products in ecDNA.** For finding NGS reads harbouring *Onsen* recombination junctions in PCR-free DNA sequencing, we first generated a virtual library of fragments containing all possible recombination products between the eight *Onsen* members. To obtain this library, we probed with a sliding-window approach the consensus sequence arising from a multiple sequence alignment between all *Onsen* members. We collapsed this consensus into a regular expression containing a wildcard character at any position at which at least one *Onsen* member showed a polymorphism differentiating them, in the form of ambiguities and indels. In each 145 bp window that we shifted by 1 nucleotide at each time, we generated *n* fragments, where *n* is the number of all possible combinations between all different polymorphisms occurring at that window. By perfect string matching, this library was purged of duplicates and sequences occurring in *Arabidopsis* TAIR10 assembly, thus recovering 501491 unique fragments which do not exist in the *Arabidopsis* genome and represent all potential recombination-like-junctions in 145 bp windows. In the next step, we performed perfect string matching of each library fragment with experimentally obtained NGS reads, to reveal the number of reads harbouring recombination-like-junctions in the experimental samples.

A second strategy to detect the presence of recombinant ecDNA was based on finding NGS recombinant molecules in heat-stressed samples. First, we used perfect string matching to recover NGS reads starting exactly at the beginning of *Onsen* LTR, which is diagnostic for linear ecDNA. This was achieved by retrieving reads bearing the 5′ LTR start sequence shared across the *Onsen* family but ruling out those displaying extra bases upstream; thus distinguishing blunt-ended ecDNA from chromosomal members or any potential integrated new copies. We then retrieved the read mates but excluded any read pair having sequences with mismatches to known *Onsen* sequences; thus obtaining only perfect matching read pairs from NGS molecules reflecting ecDNA. Finally, by perfect string matching, we assessed the *Onsen* parent-of-origin in each 5′ and 3′ pair-read as shown in Fig. 4.

**Assessment of NGS errors.** To examine whether and to which extent our strategy of detecting recombination events in ecDNA may be influenced by errors associated with NGS, we used two approaches.

The first "probabilistic" approach modelled in a mathematical formula the occurrence likelihoods of random NGS errors resulting in a recombination-like-read. The following likelihoods were incorporated in the formula: (i) $P_{error}$ (x) denoting the likelihood of a random nucleotide swap occurring in an otherwise perfect sequence of 150 bp read x, in "read" units (hence assuming all reads have 150 bp length); (ii) $P_{position}$ (x) denoting the likelihood that this error will occur at a single particular position within the read x; and (iii) $P_{base}$ (x) denoting the likelihood that this error will randomly result in a particular base required to generate a recombination-like-polymorphism (assuming each A T C G base change is equally likely to occur). The computed formula was as follows:

$r_{expected} = n * l * P_{error} (x) * P_{position} (x) * P_{base} (x)$, where $r_{expected}$ denotes the expected number of reads deceptively showing recombination-like events due to NGS random errors, *n* denotes the number of reads mapping to *Onsen*, and *l* denotes the minimum number of positions at which errors occur to generate a recombination-like-read. For our estimation, we chose the parameter set: $n = 1000$ *Onsen* mapping reads, $l = 2$ (assuming the minimum of two polymorphisms that will result in a recombination-like-read between two parental templates), $P_{error}$ (x) = 0.1415 (reflecting an error rate of 1 mismatch per 1060 bp, or 1 mismatch per ~7 reads of 150 bp; this error rate was estimated from NGS data comparing the rate of non-perfect-matching to perfect-matching reads mapping to *Onsen* across samples), $P_{position}$ (x) = 0.0066 (1 out of 150 bp), and $P_{base}$ (x) = 0.3333 (1 out of 3 possible bases). An implicit assumption is that all likelihoods $P_{error}$ (x), $P_{position}$ (x), and $P_{base}$ (x) have to occur independently. As a result, we obtained $r_{expected} = 0.62$ NGS reads/1000 mapping to *Onsen* deceptively showing recombination events due to NGS random errors.

To support this conclusion, we developed a second "permutation" approach based on generating a randomly mutated set of synthetic reads. For each permutation run we searched for recombination-reads using the previously described library of 501,491 unique 145 bp fragments. For this, we first generated a universe of all possible 150 bp synthetic reads arising from all *Onsen* sequences in a sliding-window approach at 1

nucleotide resolution, purging out repeated reads (this amounted to 48,038 unique 150 bp synthetic reads). We then randomly draw reads (with replacement) from this universe and randomly mutated the synthetic reads with probability $P_{error}$ (x) as described before. In order to accurately model the occurrence of NGS reads with recombination arising from errors for each particular DNA sequencing sample, the total number of randomly mutated synthetic reads matched the observed NGS reads mapping to *Onsen* in the actual experiments, but only reads containing mismatches were considered (i.e., non-perfect-matching reads representing sequences with both sequence errors and recombination events). The number of NGS reads with mismatches were assessed by string matching: non-perfect-matching reads in wild-type control sample were 3241 out of a total 22,105 mapping *Onsen*, in wild-type heat-stressed sample 10,332 out of a total 70,177, in *nrpd1-3* control sample 3235 out of a total 18,063, and in *nrpd1-3* heat-stressed sample 64,893 out of a total 382,403. Subsequently, we performed perfect string matching of each 145 bp library fragment with the randomly mutated synthetic reads. To model the number of reads harbouring recombination junctions in each sample, we repeated the aforementioned procedure 500 times for each modelled sample individually, thus retrieving 500 independent and randomly chosen sets of recombination-like-reads arising from modelled NGS error. The mean and maximum number of observed synthetic reads with apparent recombination junctions recorded after 500 permutations can be considered an estimation of the expected number of NGS reads deceptively appearing as recombination junctions but actually arising from NGS errors, and can be directly compared with the actual observation from the NGS experiment in each resequenced sample when using the same library of 145 bp fragments. In all modelled samples, the mean number of observed synthetic reads with apparent recombination junctions was between 0.66 and 0.80 synthetic reads/1000 mapping to *Onsen*, demonstrating that both "probabilistic" and "permutation" approaches point to very similar expected numbers of recombination-reads that can be explained by NGS errors alone. Importantly, the highest maximum number of observed synthetic reads with apparent recombination events recorded after 500 permutations was 1.55 synthetic reads/1000 mapping to *Onsen*, a figure that is still several times below the value observed in the true heat-stressed experiments (empirical observations when applying the virtual library to the actual resequenced samples were 0.36 and 0.11 NGS reads/1000 mapping to *Onsen* in control wild-type and *nrpd1-3*, respectively. In heat-stressed samples the analogous values were 7.37 and 7.55 NGS reads/1000). Therefore, we conclude that NGS errors cannot account for the observed recombination events in heat-stressed samples.

**Data availability**. Raw data was deposited in ArrayExpress (www.ebi.ac.uk/arrayexpress/) under accession numbers E-MTAB-5641 and E-MTAB-5643. All other relevant data are presented in this manuscript and its Supplementary Files, or are available from the authors upon request.

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

## Acknowledgements
We would like to thank Jayne Griffiths and Carlos Bayon for technical support. D.H.S. would also like to acknowledge Varodom Charoensawan, Hugo Tavares, Jeremy Gruel, Anna Gogleva, and Yassin Refahi for support in bioinformatics. This work was supported financially by EVOBREED ERC grant 322621 and Gatsby Fellowship AT3273/GLE.

## Author contributions
D.H.S., H.G. and J.P. designed the experiments. H.G. and D.H.S. performed the experiments. D.H.S., H.-G.D. and N.R.Z. analysed the data. D.H.S. and J.P. wrote the paper.

## Additional information

**Competing interests:** The authors declare no competing financial interests.

