## [Peer Review File · Nature Communications]

Reviewers' comments:

Reviewer #1 (Remarks to the Author):

Diego Sanchez and colleagues present the analysis of newly inserted LTR retrotransposons after an induced burst of a single LTR TE family. Most importantly, they find that the newly inserted TEs cannot be traced back to a single member of the original family. Instead the sequences reveal recombination between the family members, which could be addressed to recombination in ecDNA. As the authors point out, though there was already evidence (from the analysis of genomic sequences) that LTR TE would recombine this is the first study that analyzes and reports a surprisingly high frequency of LTR TE recombination in vivo and additionally suggests a mode of recombination. The high frequency might explain why TEs diversify at greater speed than nuclear DNA.

The manuscript is clear, short and precise. The study is very focused and it might have been interesting to read more on the broader implication of this. There are no major issues that I would have found and I fully support publication in Nat Comm after addressing some minor points.

Minor:

I got the impression that accumulation of sequence polymorphisms in the LTRs and in the TE body (which indicate the age) are taken as a proxy for activity. Wouldn't an important additional factor be the region in which the TE inserted?

Do the eight endogenous family members show patterns of recombination already?

The authors find 10 novel polymorphisms in the newly inserted sequence (160 kb) in total. The authors note that in contrast to the recombination this will only be a minor factor contributing to the diversity of LTR TE families. While I partly agree with this (only partly as recombination can only shuffle what is accumulated and therefore these things should not be compared directly but should be considered as two independent parts of the diversity), this is a really high value if compared to the mutation rate estimated for the "nuclear genome" (i.e. 1 mutation per haplotype genome and generation). Even though, the mutation rate for the nuclear genome is typically used to date insertion time for LTR sequence pairs. Do the authors suggest that we have to stop doing such insertion time dating?

In many studies the relatedness of members of LTR TE families are shown in trees, which are based on the sequences differences across the entire sequence. Assuming recombination being a major factor in all LTR TE families, such trees should not be real. What would the authors suggest how to display the relatedness of LTR TE families instead?

A simple explanation for the absence of recombinants of the green member (with discriminatory reads) in WT might simply be its lower expression in these plants?

0.6 0/00 to 0.8 0/00 per thousand reads => per mill values do not need "per thousand"?

Reviewer #2 (Remarks to the Author):

The paper by Sanchez and colleagues examines the frequency and distribution of recombination in a

family of LTR retrotransposons in Arabidopsis. By employing a system of heatshock combined with a mutation in the RdDM machinery (*nrpd1-3*), they are able to induce a high level of retrotransposon transcription and reinsertion, allowing high enough numbers for study of the contribution of each Onsen copy to recombination within retrotransposons and to transposition. Through a couple of parallel approaches, the authors (indirectly) conclude that recombination occurs at high frequency during retrotransposon reverse transcription in virus-like particles to form extra-chromosomal DNA (ecDNA). The pool of recombinant ecDNA is similar in diversity of origins and position of recombination events to that of new chromosomal insertions of Onsen in progeny plants. In terms of experimental approach, the evidence for insertion of new recombinant copies of Onsen comes only from the progeny of a single *nrpd1-3* plant. Although this result is supported by a high frequency of recombinant ecDNA molecules, given that this plant could be an outlier, it may strengthen the paper to also look at new Onsen insertions in progeny of further heat-shocked *nrpd1-3* plants. This paper is novel in addressing how recombinant retrotransposons likely arise, a topic not extensively studied thus far. This paper is therefore likely to be of interest to researchers studying transposable element biology and genome evolution as the authors conclude that transposable element diversity largely arises during transposable element transposition. I certainly found the manuscript a very interesting read.

The experimental approach is well thought out; however, I think the results could be presented in a more accessible way through some minor revisions:

1. The abstract could be expanded by an extra sentence at the end to reflect on the wider significance of the results for plant genome/transposable element evolution.
2. Page 2, second paragraph, 'It has been shown that particular steps of the transposition cycle of LTR retrotransposons are analogous to the infection cycles of retroviruses' – it might be useful to expand on what you mean here for readers not familiar with infection cycles of retroviruses. Likewise for the start of the next sentence ('in contrast to the well-documented genetic outcome of retroviral mix-infections'). Also, reference to key papers from Anna Marie Skalka's group from the early 1980's on recombination in retroviruses may be appropriate.
3. Page 6, second paragraph, 'Probabilistic and permutation models...' – here false discovery rates for recombination in ecDNA are given as 0.6‰ to 0.8‰ per thousand reads. Should this be 0.6 to 0.8 reads per thousand? Also applies to further down with 7.37‰ to 7.55‰. It would also be helpful if numbers were included for the control plants.
4. When discussing the frequency of recombination between Onsen copies, it might be useful to use the relative RNA levels from each copy (as shown in Figure 1B) to model how frequently each combination of RNAs from the same or different Onsen copies would be co-packaged into VLPs (assuming two RNAs per VLP and random assortment). This data could then be compared to the actual data from ecDNA and new insertions to see whether recombination is entirely random and if transposition from younger members is statistically significantly higher than predicted from transcript levels.
5. Page seven, third paragraph. Could be clearer on what is meant by 'the transgenic supply of marked elements'. Perhaps the authors could expand this phrase to clarify what previous evidence there is for recombination in retrotransposons and that this experimental evidence comes from yeast.
6. Figure 1: In panel A, it may be helpful to order the retrotransposons by percentage LTR identity rather than by chromosomal position as it would make the relationship between the retrotransposons

clear. Also, I think it is more normal in the field to express NGS read counts as number of reads per million rather than as counts per library size.

7. Please check colours in Figures 2 and 4 – should it be red, yellow and blue that have identical LTRs or red, yellow and grey? Admittedly, both figures could be correctly coloured if R/U5s are identical between red, yellow and blue while U3s are identical between red, yellow and grey, although from Figure 1, it would appear that red, yellow, blue, purple and grey all have identical R/U5s. Perhaps including a sequence alignment of the LTRs from each retrotransposon in the extended data (with identifying polymorphisms indicated) would allow the reader to more fully interpret the figures.

8. The retrotransposons are shown in an apparently arbitrary order in Figure 2. It might be helpful to order the retrotransposons according to parent. i.e. group 5, 27 and 32 together.

9. In the supplementary materials and methods, at what time point after heatshock were the transcriptomes and whole genome/ecDNA libraries generated?

10. Extended Data Table 2 – perhaps the data in the third column could be in the same font size as the previous columns so that all primers for a given Onsen copy align.

11. Perhaps the author could consider including a modified version of Extended Data Figure 1 to show how recombination may occur during replication.

12. The manuscript would benefit from proofreading by a native English speaker. Although the meaning is clear, there are a few grammatical errors that could be corrected.

Reviewer #3 (Remarks to the Author):

This manuscript details experimental evidence for high rates of recombination between active retrotransposons in a single generation in Arabidopsis. Because such recombination has important implications for the evolution of these repetitive elements (which often make up a majority of plant genomes) these results are of great interest to those seeking to understand the evolution of these genomes. The data suggests that during periods of high levels of activity, retroelements undergo a form of sex, which would expect to diversify families of related elements more quickly than one would expect. The degree of similarity between the elements is of some concern, but I feel that the authors have made a convincing case that they can be told apart (although they must have had to mask a lot of sequences that couldn't be mapped to individual elements). I have some concerns about quantification, since I see no evidence for an internal control. Would it have made sense to use qRT-PCR, along with the RNAseq data for a gene that is known to be up-regulated under heat stress? It was also unclear as to how new insertions were distinguished from ecDNA in the DNA sequencing experiments. Given the long experience of this group in analyzing ecDNA, I'm confident that the authors can do so, but it was not immediately apparent from the text. Finally, given the results provided here, one would expect that the authors would have more to say about how we have interpreted (or misinterpreted) the age of retroelements in the past. A core assumption has been that divergence between LTRs is an extremely accurate way to date the age of a given insertion. However, if a significant number of new insertions during transposon bursts are recombinant, don't we have to seriously re-think our dating of elements? In light of this it would be fascinating to re-evaluate the data presented in reference 18, as one would expect to find ample evidence for Retroelement recombination in that data set among the "private" insertions. Indeed, one would think that that data would make it possible to actually test the hypothesis that recombination is a common feature of

transposition bursts. Finally, it would have been nice to have been provided for new insertions in order to distinguish between somatic events and germinally transmitted events.

Major Comments:

Page 3. I do have some concerns about the use of a PCR-amplified RNAseq data set mapping to highly similar elements. Would it have been possible to normalize by comparing the read number data with read number data from a control gene that had also been subject to qRT-PCR?

Page 6: "We mapped 7.37 0/00 and 7.55 0/00 NGS reads matching recombination products in the wild type and nrpd1-3". Where is this data?

Page 4: "These data confirmed previous reports that heat stress induces the accumulation of ecDNA of Onsen in both genotypes" How are ecDNA and newly integrated elements distinguished here?

Minor Comments:

Page 2: "The remaining five family members are old..." I might be a good idea to indicate here which elements have intact ORFs.

Page 3: "especially the nrpd1- 3 mutant (Figure 1B)" should be especially in the nrpd1- 3 mutant (Figure 1B)"

In Supplemental M&M: "Several custom-made workflows were developed in-house for various data manipulations and analysis, using SAMtools (Li et al., 2009), BEDtools (Quinlan and Hall, 2010) and Python scripts" Methods should be provided such that a qualified expert can replicate the experiment. That isn't possible unless these work flows are provided as supplemental data.

REFERENCE: NCOMMS-17-12604-T

Answers to Reviewers' comments:

Reviewer #1 (Remarks to the Author):

“Diego Sanchez and colleagues present the analysis of newly inserted LTR retrotransposons after an induced burst of a single LTR TE family. Most importantly, they find that the newly inserted TEs cannot be traced back to a single member of the original family. Instead the sequences reveal recombination between the family members, which could be addressed to recombination in ecDNA. As the authors point out, though there was already evidence (from the analysis of genomic sequences) that LTR TE would recombine this is the first study that analyzes and reports a surprisingly high frequency of LTR TE recombination in vivo and additionally suggests a mode of recombination. The high frequency might explain why TEs diversify at greater speed than nuclear DNA.

The manuscript is clear, short and precise. The study is very focused and it might have been interesting to read more on the broader implication of this. There are no major issues that I would have found and I fully support publication in Nat Comm after addressing some minor points.”

Thank you.

Minor:

“I got the impression that accumulation of sequence polymorphisms in the LTRs and in the TE body (which indicate the age) are taken as a proxy for activity. Wouldn't an important additional factor be the region in which the TE inserted?”

It is true that gradual accumulation of sequence polymorphisms is considered as an approximation of potential TE activity. In contrast, the genomic region in which the TE is inserted is largely determined by properties of its integrase. Therefore, the insertion site is not a sufficient indication of potential TE activity.

“Do the eight endogenous family members show patterns of recombination already?”

It would be difficult to assess historical recombination events within the *Onsen* family due to the low number of sequence polymorphisms. However, results of the “real time” recombination (presented in Figures 2 and 3) do not reveal strong preferences for particular “pairing”. It seems that the frequency by which family members enter the reverse transcription step, and thus recombination, correlates to some degree with the abundance of their transcripts. In the revised version of the manuscript, this is illustrated in a new Extended Data Figure 4 and reinforced on page 7, using the “green” member as example: *“In nrpd1-3, where transcripts of the “green” member were much more abundant (Figure 1B), this parental member also contributed significantly to the extrachromosomal recombinant LTRs we detected (Figure 4B).”*

“The authors find 10 novel polymorphisms in the newly inserted sequence (160 kb) in total. The authors note that in contrast to the recombination this will only be a minor factor contributing to

the diversity of LTR TE families. While I partly agree with this (only partly as recombination can only shuffle what is accumulated and therefore these things should not be compared directly but should be considered as two independent parts of the diversity), this is a really high value if compared to the mutation rate estimated for the “nuclear genome” (i.e. 1 mutation per haplotype genome and generation). “

We agree with this comment and we have modified the statement on page 6, which now reads: “In approximately 160 kb of sequence assigned to new insertions, we detected 10 sequence polymorphisms not present in the parental members (Extended Data Table 1). These most likely represent errors during reverse transcription, which are a known source of novel mutations in retroelements⁹. However, this sequence variability is rather marginal compared to the observed sequence diversity derived from the recombination step during a retrotransposition burst, which reshuffles mutations accumulated during years of passive maintenance of retroelements within the host chromosomes.”

*The added citation is reference number nine: Kumar, A. & Bennetzen, J. L. Plant retrotransposons. *Annual review of genetics* 33, 479-532, doi:10.1146/annurev.genet.33.1.479 (1999).*

Consequently, we also deleted the statement: “Therefore, the replication/transposition cycle of the Onsen family occurs with remarkably high fidelity”.

“Even though, the mutation rate for the nuclear genome is typically used to date insertion time for LTR sequence pairs. Do the authors suggest that we have to stop doing such insertion time dating?”

We do not suggest this. LTRs always become identical due to the replication mechanism of LTR-TEs (even if they have undergone recombination). Thus, they provide the correct starting point for dating by mutation rate.

“In many studies the relatedness of members of LTR TE families are shown in trees, which are based on the sequences differences across the entire sequence. Assuming recombination being a major factor in all LTR TE families, such trees should not be real. What would the authors suggest how to display the relatedness of LTR TE families instead?”

This is a very important and valid point. As cited and discussed in the manuscript (page 7), isolated historical recombination events of LTR-TEs were deduced from their “family trees”. Now, given the extremely high frequency of recombination during bursts of family transposition, the “family trees” probably need to be adjusted for this new variable. It would be interesting to learn from evolutionary biologists how they intend to deal with this issue.

“A simple explanation for the absence of recombinants of the green member (with discriminatory reads) in WT might simply be its lower expression in these plants?”

Yes, this is correct and was suggested in the text on page 7.

“0.6 0/00 to 0.8 0/00 per thousand reads => per mill values do not need “per thousand?”

Thank you for noticing this. We now correctly state on page 6: “...0.6 to 0.8 recombination events per thousand reads mapping to Onsen may be due to random errors of NGS present in our data sets”.

Reviewer #2 (Remarks to the Author):

“The paper by Sanchez and colleagues examines the frequency and distribution of recombination in a family of LTR retrotransposons in Arabidopsis. By employing a system of heatshock combined with a mutation in the RdDM machinery (nrpd1-3), they are able to induce a high level of retrotransposon transcription and reinsertion, allowing high enough numbers for study of the contribution of each Onsen copy to recombination within retrotransposons and to transposition. Through a couple of parallel approaches, the authors (indirectly) conclude that recombination occurs at high frequency during retrotransposon reverse transcription in virus-like particles to form extra-chromosomal DNA (ecDNA). The pool of recombinant ecDNA is similar in diversity of origins and position of recombination events to that of new chromosomal insertions of Onsen in progeny plants. In terms of experimental approach, the evidence for insertion of new recombinant copies of Onsen comes only from the progeny of a single nrpd1-3 plant. Although this result is supported by a high frequency of recombinant ecDNA molecules, given that this plant could be an outlier, it may strengthen the paper to also look at new Onsen insertions in progeny of further heat-shocked nrpd1-3 plants.”

This is a very valid point and the revised manuscript is supplemented by LTR reconstructions of an independent resequenced line, which gave results comparable to those in Figure 2. We have referred to this new Extended Data Figure 3A on page 5 of the main text. This now reads: *“Additional reconstruction of LTR sequences from new Onsen insertions in an independent nrpd1-3 plant produced comparable results (Extended Data Figure 3).”*

Also, we included partial sequences of new Onsen insertions from a distinct set of plants in which the high frequency recombination was first discovered (now added as Extended Data Figure 3B). We did not present these data initially because subsequent sequencing was more complete and covered the full lengths of new recombinant Onsen copies. To avoid interrupting the flow of the main text of the manuscript, we have referred to this new Extended Data Figure 3B in the Supplementary Material and Methods (page 2).

“This paper is novel in addressing how recombinant retrotransposons likely arise, a topic not extensively studied thus far. This paper is therefore likely to be of interest to researchers studying transposable element biology and genome evolution as the authors conclude that transposable element diversity largely arises during transposable element transposition. I certainly found the manuscript a very interesting read.”

Thank you.

“The experimental approach is well thought out; however, I think the results could be presented in a more accessible way through some minor revisions:

1. *The abstract could be expanded by an extra sentence at the end to reflect on the wider significance of the results for plant genome/transposable element evolution.*

As suggested, the abstract is now supplemented by the final sentence: “Our observations provide an explanation for the reported high rates of sequence diversification in LTR retrotransposons.”

“2. Page 2, second paragraph, ‘It has been shown that particular steps of the transposition cycle of LTR retrotransposons are analogous to the infection cycles of retroviruses’ – it might be useful to

expand on what you mean here for readers not familiarly with infection cycles of retroviruses. Likewise for the start of the next sentence ('in contrast to the well-documented genetic outcome of retroviral mix-infections'). Also, reference to key papers from Anna Marie Skalka's group from the early 1980's on recombination in retroviruses may be appropriate."

The requested information and the citation have been added. The new text on page 2 now reads: *"It has been shown that particular steps of the transposition cycle of LTR retrotransposons, such as reverse transcription and chromosomal integration of DNA, are analogous to the infection cycles of retroviruses⁵. However, in contrast to the well-documented formation of recombinant strains during retroviral mix-infections³, the genetic consequences of the transposition bursts of an entire family of LTR retroelements, consisting of young and old members, are unknown."*

The added citation is reference number five: Ju, G. & Skalka, A. M. Nucleotide sequence analysis of the long terminal repeat (LTR) of avian retroviruses: structural similarities with transposable elements. *Cell* 22, 379-386 (1980).

"3. Page 6, second paragraph, 'Probabilistic and permutation models...' – here false discovery rates for recombination in ecDNA are given as 0.6‰ to 0.8‰ per thousand reads. Should this be 0.6 to 0.8 reads per thousand? Also applies to further down with 7.37‰ to 7.55‰. It would also be helpful if onnumbers were included for the control plants. "

We have now corrected this statement on page 6: *"Probabilistic and permutation models predicted that approximately 0.6 to 0.8 recombination events per thousand reads mapping to Onsen may be due to random errors of NGS present in our data"*. The corrections were applied in subsequent statements and also in the Supplementary Material and Methods.

As suggested, the values for the control plants have also been added. The modified Supplementary Material and Methods on page 6 now reads: *"empirical observations when applying the virtual library to the actual resequenced samples where 0.36 and 0.11 NGS reads/1000 mapping to Onsen in control wild type and nrpd1-3, respectively. In heat stress samples the analogous values were 7.37 and 7.55 NGS reads/1000"*.

"4. When discussing the frequency of recombination between Onsen copies, it might be useful to use the relative RNA levels from each copy (as shown in Figure 1B) to model how frequently each combination of RNAs from the same or different Onsen copies would be co-packaged into VLPs (assuming two RNAs per VLP and random assortment). This data could then be compared to the actual data from ecDNA and new insertions to see whether recombination is entirely random and if transposition from younger members is statistically significantly higher than predicted from transcript levels."

This is an interesting suggestion. However, the relatively low number of available and fully sequenced new insertions so far precludes robust statistical modelling. Nevertheless, we have now included a direct comparison of relative mRNA levels to the relative composition of *Onsen* neo-insertions (Extended Data Figure 4). Consequently, we modified the manuscript on page 5, which now reads: *"However, in the recombination process, sequences of old family members were habitually incorporated into progeny insertions (as revealed by analysis of the relative contributions of parental Onsen family members to newly integrated copies; Extended Data Figure 4). This created recombinational diversity among the progeny of the Onsen family"*.

Additionally, we modified the Supplementary Materials and Methods to explain the methodologies of this analysis. In this section, we added a new sub-section "Assessment of *Onsen* member contributions to new insertions", which reads: *"To assess the relative contributions of each Onsen*

family member to the sequences of the 32 new insertions, we devised a scoring system that quantified each recombination event in terms of the contribution of a particular parental Onsen to the new recombined insertion. We assigned up to 12 points per new insertion. Where only one parental member contributed to a new insertion, we assigned 12 points. We assigned each member 6 points where two different parental family members formed a new insertion through a recombination event; where three members contributed, each received a score of 4 points, and so on. We then calculated the total “contribution scores” for each Onsen member over all 32 new insertions (32 scores per member). Finally, we calculated and visualized the relative frequencies of these total contribution scores to determine whether they matched the relative levels of transcripts assigned to each parental Onsen (Extended Data Figure 4).”

“5. Page seven, third paragraph. Could be clearer on what is meant by ‘the transgenic supply of marked elements’. Perhaps the authors could expand this phrase to clarify what previous evidence there is for recombination in retrotransposons and that this experimental evidence comes from yeast.”

The sentence was expanded accordingly and page 8 now reads: “However, the only experimental support so far has come from the transgenic supply of marked yeast Ty elements^{17,18}”

“6. Figure 1: In panel A, it may be helpful to order the retrotransposons by percentage LTR identity rather than by chromosomal position as it would make the relationship between the retrotransposons clear. “

We would prefer to order by chromosomal position, since this order was used in the first publication describing the Onsen family (Ito et al. 2011). The members were unfortunately “renamed” later (Cavrak VV et al. 2014), making comparison of the data in the two manuscripts difficult. Further reshuffling would only increase the probability of confusion.

“Also, I think it is more normal in the field to express NGS read counts as number of reads per million rather than as counts per library size.”

Thank you for pointing this out. Unfortunately, the Y-axis of Figure 1B was mislabelled, because “reads per million” were actually depicted. We are sorry for this mistake, which we have now corrected, also in Extended Data Figure 2.

“7. Please check colours in Figures 2 and 4 – should it be red, yellow and blue that have identical LTRs or red, yellow and grey? Admittedly, both figures could be correctly coloured if R/U5s are identical between red, yellow and blue while U3s are identical between red, yellow and grey, although from Figure 1, it would appear that red, yellow, blue, purple and grey all have identical R/U5s. Perhaps including a sequence alignment of the LTRs from each retrotransposon in the extended data (with identifying polymorphisms indicated) would allow the reader to more fully interpret the figures.”

As depicted in Figure 1 and clarified in the main text, Onsen members with identical LTRs are red, yellow and grey. Figures 2 and 4 depict different types of data. The former represent sequences of reconstructed complete chromosomal LTRs, from which 5’LTRs and 3’LTRs can be distinguished in older Onsen members (as depicted by horizontal or vertical bars). However, Figure 4 involves 5’ and 3’ NGS reads from the same sequenced molecules of LTRs in ecDNA, which may not present sequence polymorphisms that allow discrimination of the 5’ or 3’ LTR origin from a particular older

member. Hence, in these cases, Figure 4 areas have been coloured without horizontal or vertical bars.

To avoid this misunderstanding, we have modified the legend to Figure 4. The final sentences of it on page 10 now reads: *“(B) Discriminatory pair-end reads reveal the occurrence of recombinant LTRs. Colour code as in Figure 1A. Horizontal or vertical bars are present only in those cases where NGS reads matching old Onsen members contained polymorphisms that allowed discrimination of 5’ and 3’ LTRs”*.

“8. The retrotransposons are shown in an apparently arbitrary order in Figure 2. It might be helpful to order the retrotransposons according to parent. i.e. group 5, 27 and 32 together.”

Since the numbers for a given insertion displayed in Figure 2 correspond to the numbers in Figure 3 of insertions analysed in more detail, we would prefer not to reshuffle these two figures.

“9. In the supplementary materials and methods, at what time point after heatshock were the transcriptomes and whole genome/ecDNA libraries generated?”

As suggested, this information has been added on page 1 of the revised Supplementary Material and Methods; the text now reads: *“Pools of 50-70 seedlings from independent biological replicates were collected directly after heat stress and control treatments.”*

“10. Extended Data Table 2 – perhaps the data in the third column could be in the same font size as the previous columns so that all primers for a given Onsen copy align.”

This was created by a mis-conversion in the manuscript tracking system and has now been corrected by new formatting of Extended Data Table 2.

“11. Perhaps the author could consider including a modified version of Extended Data Figure 1 to show how recombination may occur during replication. “

Extended Data Figure 1 is already very busy and we are afraid that duplication of the number of molecules to illustrate template switching/recombination would make this figure too complicated. We would prefer to avoid this.

“12. The manuscript would benefit from proofreading by a native English speaker. Although the meaning is clear, there are a few grammatical errors that could be corrected.”

The new version of the manuscript has now been corrected by a native English speaker.

Reviewer #3 (Remarks to the Author):

“This manuscript details experimental evidence for high rates of recombination between active retrotransposons in a single generation in Arabidopsis. Because such recombination has important implications for the evolution of these repetitive elements (which often make up a majority of plant genomes) these results are of great interest to those seeking to understand the evolution of these genomes. The data suggests that during periods of high levels of activity, retroelements undergo a form of sex, which would expect to diversify families of related elements more quickly than one would expect. “

Thank you.

“The degree of similarity between the elements is of some concern, but I feel that the authors have made a convincing case that they can be told apart (although they must have had to mask a lot of sequences that couldn’t be mapped to individual elements). I have some concerns about quantification, since I see no evidence for an internal control. Would it have made sense to use qRT-PCR, along with the RNAseq data for a gene that is known to be up-regulated under heat stress? “

Importantly, we used perfect string matching instead of standard mapping tools (described in Supplementary Material and Methods). Therefore, elements can be distinguished very well by informative SNPs/indels. For quantification, the RNA-seq data were normalized to library size and thus equalized across independent samples; they were also compared to samples of non-stress controls. These methods proved reliable given that independent biological replicates displayed similar data (Figure 1 and Extended Data Figure 2).

However, we agree that the use of genes for validation that are upregulated under heat stress would be of advantage. Therefore, using our strategy of specific sequence addresses, we have examined a few housekeeping and heat responsive genes. This was compared with standard determination of transcript levels (RPKM) from transcriptomic data. The two methods correlated very well across genes in both control and heat-stress wild type samples, demonstrating that the “address” strategy determines transcript levels just as well as the standard transcriptomic approach. These data have been added as a new Extended Data Figure 5. The Supplementary Material and Methods has been modified accordingly on page 3 and now reads: *“The validation of this approach for estimating transcript levels was corroborated by correlation with standard transcript level determination (RPKM), across both wild type transcriptome data and case examples of housekeeping and heat responsive genes (Extended Data Figure 5)”*.

“It was also unclear as to how new insertions were distinguished from ecDNA in the DNA sequencing experiments. Given the long experience of this group in analyzing ecDNA, I’m confident that the authors can do so, but it was not immediately apparent from the text. “

Since the sequencing of new insertions was performed on DNA isolated from progeny plants that were not heat stressed, ecDNA was not produced. This point has now been clarified in the modified Supplementary Material and Methods on page 3: *“The sequencing for finding new insertions was performed on DNA isolated from control progeny of plants not subjected to heat stress and, therefore, ecDNA was absent.”*

In the case of sequencing ecDNA, this was distinguished by the absence of flanking chromosomal DNA linked to LTR's ends. This we have now explained better in the modified Supplementary Material and Methods section on page 4, which now reads: *"A second strategy to detect the presence of recombinant ecDNA was based on finding NGS recombinant molecules in heat stressed samples. First, we used perfect string matching to recover NGS reads starting exactly at the beginning of Onsen LTR, which is diagnostic for linear ecDNA. This was achieved by retrieving reads bearing the 5' LTR start sequence shared across the Onsen family but ruling out those displaying extra bases upstream; thus distinguishing blunt-ended ecDNA from chromosomal members or any potential integrated new copies"*.

"Finally, given the results provided here, one would expect that the authors would have more to say about how we have interpreted (or misinterpreted) the age of retroelements in the past. A core assumption has been that divergence between LTRs is an extremely accurate way to date the age of a given insertion. However, if a significant number of new insertions during transposon bursts are recombinant, don't we have to seriously re-think our dating of elements? In light of this it would be fascinating to re-evaluate the data presented in reference 18, as one would expect to find ample evidence for Retroelement recombination in that data set among the "private" insertions. Indeed, one would think that that data would make it possible to actually test the hypothesis that recombination is a common feature of transposition bursts. "

These are very interesting points for future research. Regarding dating of retroelements since their last transposition, we believe that divergence between LTRs of the same element is still an accurate way to estimate age. Despite recombination, LTRs tend to become identical due to the replication mechanism of LTR-TEs and this is also valid for the recombinant LTRs. Considering reference 18, these data are not suitable for tracing recombinant products, since the focus was on new insertion sites and not on the structure of the elements themselves. However, in the future, the emphasis may change with long sequence reads and the tracing of recombination events of retroelements during evolution of host genomes could result in surprising findings.

"Finally, it would have been nice to have been provided for new insertions in order to distinguish between somatic events and germinally transmitted events."

This is a very valid point; however, due to the lack of phenotypes for somatic events (e.g. visible sectors) we could only trace germline transmissible insertions. We believe that the transgenerational events traced here are most crucial for shaping the evolution of genomes.

"Major Comments:

Page 3. I do have some concerns about the use of a PCR-amplified RNAseq data set mapping to highly similar elements. Would it have been possible to normalize by comparing the read number data with read number data from a control gene that had also been subject to qRT-PCR?"

We have now used case examples of housekeeping and heat responsive genes for validation of our strategy. These data have been added as a new Extended Data Figure 5. The Supplementary Material and Methods section was modified accordingly, and page 3 now reads: *"The validation of this approach for estimating transcript levels was corroborated by correlation with standard transcript level determination (RPKM), across both wild type transcriptome data and case examples of housekeeping and heat responsive genes (Extended Data Figure 5)."*

“Page 6: “We mapped 7.37 0/00 and 7.55 0/00 NGS reads matching recombination products in the wild type and nrpd1-3”. Where is this data?”

These data were obtained by running the workflow of analysis described in Supplementary Material and Methods under the section “Assessment of NGS errors”, when applying the virtual library of synthetic 145 bp recombination products to the actual resequencing data in heat stressed wild type and *nrpd1-3* plants. We have now provided the scripts of these workflows at <https://github.com/diego hernansanchez> (clarified on page 2 of the new Supplementary Material and Methods).

To avoid any misunderstanding, we have modified the text on page 6, which now reads: *“Probabilistic and permutation models predicted that approximately 0.6 to 0.8 recombination events per thousand reads mapping to Onsen may be due to random errors of NGS present in our data sets (Supplementary Material and Methods). Application of the search strategy of the virtual library to the actual sequencing data gave frequencies of putative recombination junctions below the values expected in samples from control plants (Supplementary Material and Methods). In contrast, the frequencies of sequences indicative of recombination were significantly elevated in heat stressed plants. We mapped 7.37 and 7.55 NGS reads matching recombination products in the wild type and nrpd1-3, respectively, normalized per thousand reads mapping to Onsen.”*

“Page 4: “These data confirmed previous reports that heat stress induces the accumulation of ecDNA of Onsen in both genotypes” How are ecDNA and newly integrated elements distinguished here? “

This point has been addressed above.

“Minor Comments:

Page 2: “The remaining five family members are old...” I might be a good idea to indicate here which elements have intact ORFs. “

The legend of Figure 1 has now been modified to clarify which old *Onsen* members have an intact coding-sequence. The added text in the legend to Figure 1 in page 9 now reads: *“Old members AT1G48710/“violet” and AT1G58140/“green” display complete central CDS.”*

“Page 3: “especially the nrpd1- 3 mutant (Figure 1B)” should be especially in the nrpd1- 3 mutant (Figure 1B)””

The statement has been corrected.

“In Supplemental M&M: “Several custom-made workflows were developed in-house for various data manipulations and analysis, using SAMtools (Li et al., 2009), BEDtools (Quinlan and Hall, 2010) and Python scripts” Methods should be provided such that a qualified expert can replicate the experiment. That isn’t possible unless these work flows are provided as supplemental data.”

These workflows are now available at <https://github.com/diego hernansanchez/>. This is also clarified on page 2 of the modified Supplementary Material and Methods.

REVIEWERS' COMMENTS:

Reviewer #1 (Remarks to the Author):

I would like to thank the authors for carefully addressing my concerns. All of them have been answered convincingly, and I fully support publication of the manuscript in Nat Comm.

Reviewer #2 (Remarks to the Author):

The authors have addressed all concerns raised. There are just a couple of extremely minor points below with regards to the revisions, however I now fully support publication of this study.

1. In extended data figure 4A, the second bar should presumably be white rather than grey.

2. Phrasing is a little awkward on page 3 where the word mutant is used twice in the sentence: 'transposition results in new chromosomal insertions in mutant progeny of heat stressed Onsen mutants deficient in the biogenesis of small interfering RNAs'.

Reviewer #3 (Remarks to the Author):

The authors have satisfied my concerns.

REFERENCE: NCOMMS-17-12604A

Answers to Reviewers' comments:

Reviewer #2 (Remarks to the Author):

"In extended data figure 4A, the second bar should presumably be white rather than grey."

Thank you for pointing this out; the current Supplementary Fig. 4a has been corrected.

"Phrasing is a little awkward on page 3 where the word mutant is used twice in the sentence: 'transposition results in new chromosomal insertions in mutant progeny of heat stressed Onsen mutants deficient in the biogenesis of small interfering RNAs'."

Thank you for noticing this editing error. The sentence has now been corrected and reads as follows: *'In contrast, transposition results in new chromosomal insertions in the progeny of heat stressed mutant plants deficient in the biogenesis of small interfering RNAs (siRNAs)'*